# Quantifying the impact of ecological memory on the dynamics of interacting communities

**Moein Khalighi**[1☯]*, **Guilhem Sommeria-Klein**[1☯], **Didier Gonze**[2], **Karoline Faust**[3], **Leo Lahti**[1]*

**1** Department of Computing, Faculty of Technology, University of Turku, Turku, Finland, **2** Unité de Chronobiologie Théorique, Faculté des Sciences CP 231, Université Libre de Bruxelles, Brussels, Belgium, **3** Laboratory of Molecular Bacteriology (Rega Institute), Department of Microbiology, Immunology and Transplantation, KU Leuven, Leuven, Belgium

☯ These authors contributed equally to this work.
* moein.khalighi@utu.fi (MK); leo.lahti@utu.fi (LL)

**Data Availability Statement:** The computational results for this article have been generated with MATLAB. All data and code used for running the simulations and generating the figures is available in GitHub, and accessible via the permanent

## Abstract

Ecological memory refers to the influence of past events on the response of an ecosystem to exogenous or endogenous changes. Memory has been widely recognized as a key contributor to the dynamics of ecosystems and other complex systems, yet quantitative community models often ignore memory and its implications.

Recent modeling studies have shown how interactions between community members can lead to the emergence of resilience and multistability under environmental perturbations. We demonstrate how memory can be introduced in such models using the framework of fractional calculus. We study how the dynamics of a well-characterized interaction model is affected by gradual increases in ecological memory under varying initial conditions, perturbations, and stochasticity.

Our results highlight the implications of memory on several key aspects of community dynamics. In general, memory introduces inertia into the dynamics. This favors species coexistence under perturbation, enhances system resistance to state shifts, mitigates hysteresis, and can affect system resilience both ways depending on the time scale considered. Memory also promotes long transient dynamics, such as long-standing oscillations and delayed regime shifts, and contributes to the emergence and persistence of alternative stable states. Our study highlights the fundamental role of memory in communities, and provides quantitative tools to introduce it in ecological models and analyse its impact under varying conditions.

## Author summary

An ecosystem is said to exhibit *ecological memory* when its future states do not only depend on its current state but also on its initial state and trajectory. Memory may arise through various mechanisms as organisms adapt to their environment, modify it, and accumulate biotic and abiotic material. It may also emerge from phenotypic heterogeneity at the population level. Despite its commonness in nature, ecological memory and its

Zenodo DOI: https://doi.org/10.5281/zenodo.5979561.

**Funding:** This study was funded by the Academy of Finland (URL: https://www.aka.fi/), the University of Turku (URL: https://www.utu.fi/en/research/utugs/dpt), and the European Union's Horizon 2020 research and innovation program and European Research Council (ERC) (URL: https://ec.europa.eu/programmes/horizon2020/en/home). The salary for MK was covered by the University of Turku Graduate School Doctoral Programme in Technology (UTUGS/DPT). Academy of Finland covered the salary for GSK (decisions 340314, to GSK) and for LL (decisions 295741 and 330887, to LL). The European Union's Horizon 2020 research and innovation program (grant 952914, to LL) and the Sakari Alhopuro foundation (grant 20210172, to GSK) covered partial salary for GSK. KF was supported by the European Research Council (ERC; grant 801747, to KF). The funders had no role in study design, data collection and analysis, decision to publish, or preparation of the manuscript.

**Competing interests:** The authors have declared that no competing interests exist.

potential influence on ecosystem dynamics have been so far overlooked in many applied contexts. Here, we use modeling to investigate how memory can influence the dynamics, composition, and stability landscape of communities. We incorporate long-term memory effects into a multi-species model recently introduced to investigate alternative stable states in microbial communities. We assess the impact of memory on key aspects of model behavior and further examine our findings using a model parameterized by empirical data from the human gut microbiota. Our approach for modeling long-term memory and studying its implications has the potential to improve our understanding of microbial community dynamics and ultimately our ability to predict, manipulate, and experimentally design microbial ecosystems. It could also be applied more broadly in the study of systems composed of interacting components.

## Introduction

The temporal variations observed in ecosystems arise from the interplay of complex deterministic and stochastic processes, the identification and characterization of which requires quantitative models. The empirical study of microbial communities provides an ideal source of data to inform the development of dynamical community models, since this active research area generates dense ecological time series under highly controlled experimental conditions and perturbations [1, 2]. Nevertheless, despite the recent advances in metagenomic sequencing and other high-throughput profiling technologies that are now transforming the analysis of microbial communities [3], there has been only limited success in accurately modeling and predicting the dynamics of microbial communities, even in well-controlled laboratory conditions [2, 4, 5]. This highlights the need to re-evaluate and extend the available models to better account for the various mechanisms that underlie community dynamics [2, 6–9]. One notable shortcoming of the currently popular dynamical models of microbial communities is that they ignore the role of memory, that is, they are based on the assumption that the community's future behavior solely depends on its current state, perturbations, and stochasticity.

Ecological memory is present when the community's past states and trajectories influence its dynamics over extended periods. This is a fundamental and ubiquitous aspect of natural communities, and its influence on community dynamics has been widely recognized across ecological systems [10–12]. Memory can emerge at different time scales through a number of mechanisms, including the accumulation of abiotic and biotic material characterizing past legacies of the system, adaptations to past conditions, dormancy, or spatial structure [13–18]. Thus, developing and investigating new means to incorporate memory in dynamical models of communities has the potential to yield more accurate mechanistic understanding and predictions.

Diverse approaches have been proposed to explore ecological memory, including time delays [11, 19, 20], historical effects [21], exogenous memory [12], and buffering of disturbances [22]. A stochastic framework was recently used to evaluate the length, patterns, and strength of memory in a series of ecological case studies [11]. However, none of these approaches describes long-term memory with a power-law decay of the influence of past states. The lag times of antibiotic-tolerant persister cells have been shown to be power-law distributed in bacterial populations [23], and this type of long-term memory is likely to be common in microbial communities whenever memory emerges from phenotypic heterogeneity [16, 24]. Furthermore, the impact of memory has not been systematically addressed in contemporary

studies, and specific methods have been missing for incorporating memory into standard deterministic models of microbial community dynamics.

Potential community assembly mechanisms have been recently investigated based on extensions of the generalized Lotka-Volterra (gLV) model, which provides a general modeling framework for species interactions [25–27]. The standard model has been extended by incorporating external perturbations [28] and sequencing noise [29], and to satisfy specific modeling constraints such as compositionality [30, 31]. gLV models have also been combined with Bayesian Networks for improved longitudinal predictions [32]. One goal of these modeling efforts is to understand how alternative community types reported in the human microbiome may arise, possibly in combination with external factors [33–36]. Despite the recent popularity of gLV models in microbial ecology, the impact of memory in these models has been largely ignored.

We address the above shortcomings by explicitly incorporating long-term memory effects into community interaction models using fractional calculus, which provides well-established tools for modeling memory [37, 38]. We incorporate memory into a gLV model with multiplicative species interactions that was recently used to reproduce the alternative stable states observed empirically in the human gut microbiota [25], and we use this extended model to analyze and demonstrate how memory can influence critical aspects of community dynamics. We then assess our findings by adding memory to a gLV model parameterized with experimental data [39]. Our work contributes to the growing body of quantitative techniques for studying community resistance, resilience, prolonged instability, transient dynamics, and abrupt regime shifts [40–44].

## Results

### Modeling memory

The gLV and its extensions are ordinary differential equation systems. This class of models has been commonly used to model community dynamics, but their standard formulations ignore memory effects. Here, we show how ecological memory can be included in these models using *fractional calculus*. This mathematical tool provides a principled framework for incorporating memory effects into differential equation systems (see *e.g.* [37, 38, 45]), thus allowing a systematic analysis and quantification of memory effects in commonly used dynamical models of communities.

Let us first consider a simple community with three species that tend to inhibit each other's growth (Fig 1A). We will later extend this model community to a larger number of species. To model this system, we employ a non-linear extension of the gLV model that was recently used to demonstrate possible mechanisms underlying the emergence of alternative states in a community [25]. This model describes the dynamics of a species $i$ as a function of its growth rate, death rate, and a multiplicative interaction term function of the interaction matrix between all species pairs, as described in Fig 1A. Under certain conditions, this model gives rise to a tristable community, where each stable state corresponds to the dominance of a different species. The community can shift from one stable state to another following a perturbation (Fig 1B). Such transitions can be for instance controlled by changes in the species' growth rates.

To introduce memory, we extend this model by incorporating fractional derivatives. In this extended formulation, the classical derivative operator $d/dt$ is replaced by the fractional derivative operator $\mathfrak{D}^{\mu_i}$, where $\mu_i \in (0, 1]$ is the non-integer derivative order for species $i$ (Fig 1C). The fractional derivative is defined by a convolution integral with a power-law memory kernel (see Methods). The $\mu_i$ can then be used as a tuning parameter for memory, with lower values of $\mu_i$ indicating higher levels of memory for species $i$ [37]. The *strength of memory* for species $i$

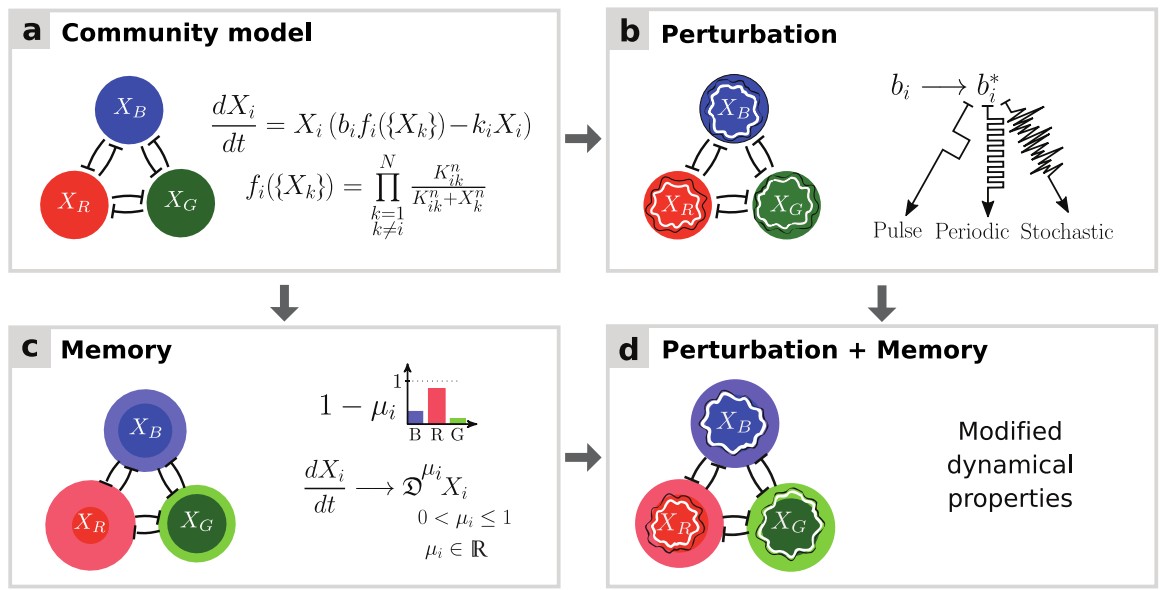

**Fig 1. Schematic illustration of a three-species community in the presence of memory and perturbations. (A)** Mutual interaction model introduced by Gonze et al. [25] to illustrate the emergence of alternative stable states in human gut microbial communities. The model describes the dynamics of species abundances $X_i$ as functions of growth rates $b_i$, death rates $k_i$, and inhibition functions $f_i$, where $K_{ij}$ and $n$ denote interaction constants and Hill coefficients, respectively. **(B)** Standard perturbations include pulse, periodic, and stochastic variation in species immigration, death, or growth rates. Such perturbations may trigger shifts between alternative states. **(C)** Memory (bolded circles) can be incorporated into dynamical models by substituting the integer-order derivatives with fractional derivatives $\mathfrak{D}^{\mu_i}$ of order $\mu_i$ (see [37] and Methods). As decreasing $\mu_i$ values correspond to increasing memory, memory is measured as $1 - \mu_i$. When all community members have the same memory ($\mu_i = \mu$ for all $i$), the system is said to have *commensurate* memory, otherwise *incommensurate*. Increasing memory changes community dynamics, in particular by introducing inertia and modifying the stability landscape around stable states. **(D)** Ecological memory can change the system dynamics under perturbations.

is measured as $1 - \mu_i$. This model includes two special cases: (i) *no memory* ($\mu_i = \mu = 1$ for all species $i$), which corresponds to the original community model with classical first-order derivatives, and (ii) *commensurate memory*, where all species have equal memory ($\mu_i = \mu \leq 1$). In contrast, the general case is referred to as *incommensurate memory*, where $\mu_i$ may be unique for each $i$, and hence the degree of memory may differ between species. We numerically solve the fractional-order model with varying values of the parameter $\mu_i$, thus inducing varying levels of memory, and use it to analyse the effect of memory on various aspects of community dynamics, in particular its response to perturbations (Fig 1D).

## Resistance and resilience to perturbation

*Resistance* refers to a system's capacity to withstand a perturbation without changing its state, while *resilience* refers to its capacity to recover to its original state after a perturbation [46, 47]. To examine the impact of ecological memory on community resistance and resilience in response to perturbations, we perturbed the system by changing the species growth rates over time. Specifically, we investigated the three-species community under *pulse* (Fig 2), *periodic* (Fig 3), and *stochastic* (Fig 4) perturbations, and analysed the impact of these three types of perturbations on community dynamics in the presence of (commensurate) memory.

Our results show that memory tends to increase resistance to perturbations by allowing the competing species' coexistence for a longer time. In the presence of memory, switches between alternative community states take place more slowly following a pulse perturbation (Fig 2A), or in some cases may be prevented entirely (Fig 2B). S1 Fig provides a further example of the

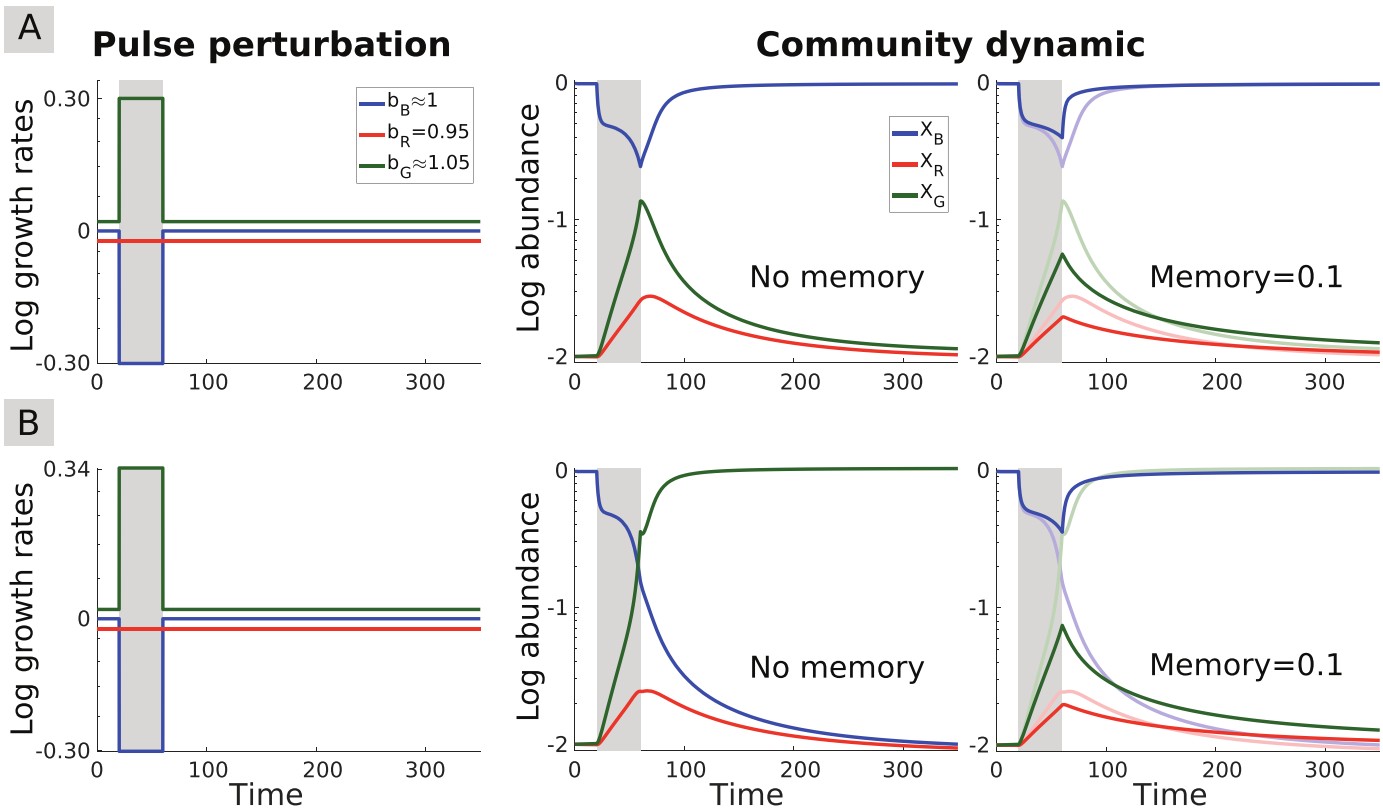

**Fig 2. Impact of commensurate memory on community resistance and resilience. (A)** A pulse perturbation is applied to the community (left panel): the growth rate of the blue species is lowered while that of the green species is simultaneously raised. The perturbation temporarily moves the community away from its initial stable state, characterized by blue species dominance (middle panel). Introducing commensurate memory (right panel) increases resistance to perturbation since the community is not displaced as far from its initial state compared to the memoryless case (shown in superimposition). The effect on resilience depends on the time scale considered: while memory initially hastens the recovery after the perturbation, it slows down the later stages of the recovery (starting around the time step 150). **(B)** A slightly stronger pulse perturbation is applied (left panel), triggering a shift toward an alternative stable state dominated by the green species (middle panel). Memory can prevent the state shift (right panel), thus increasing both resistance and resilience to perturbation.

increased resistance provided by memory in a larger, unstructured community where memory helps preserve the stable state after a pulse perturbation compared to the corresponding memoryless system.

After the perturbation has ceased, memory initially hastens the return to the original state, but then slows it down in the later stages of the recovery (Fig 2A). Thus, the impact of memory on resilience is multi-faceted: depending on the time scale considered, memory either hastens or slows down the recovery from perturbations, thus increasing or reducing resilience, respectively. Long-term memory may indeed act across several time scales owing to the slow (here power-law) decay of the influence of past states. Furthermore, in multistable systems, memory enhances resilience by promoting the persistence of the original stable state (Fig 2B).

Considering two successive pulse perturbations in opposite directions (Fig 3A) highlights another way memory can affect resilience in multistable systems. After a state shift triggered by a first perturbation, memory hastens recovery to the initial state following a second, opposite perturbation, hence increasing long-term resilience (Fig 3B). Memory can thus mitigate hysteresis, known to be a feature of several ecological systems.

In the presence of regularly alternating opposite pulse perturbations (Fig 3C), akin to those experienced by the gut microbiome or marine plankton, the community may not be able to

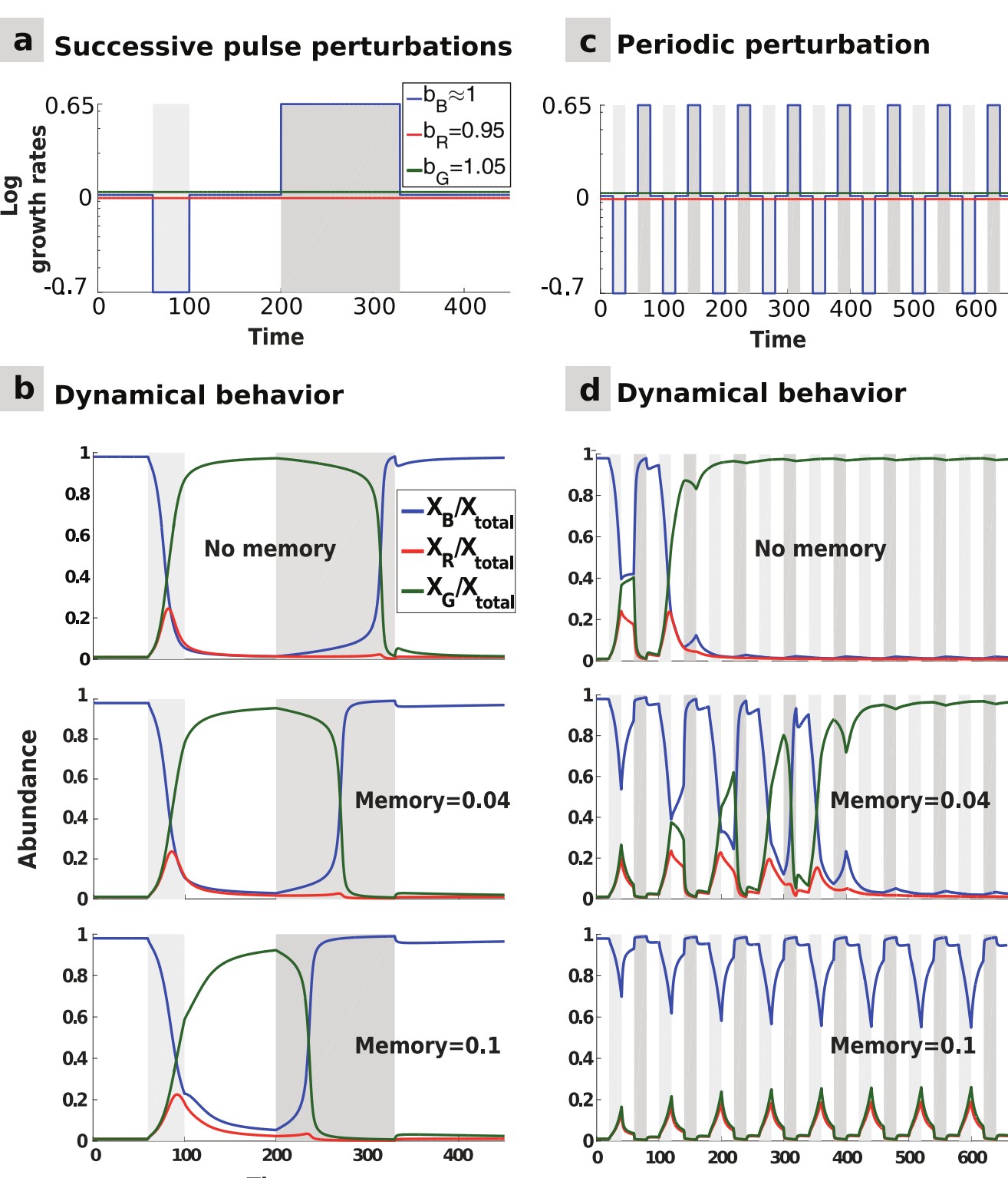

**Fig 3. Multi-pulse and periodic perturbations: Commensurate memory impact on hysteresis and transient oscillations. (A)** Two opposite pulse perturbations are applied successively: the blue species growth rate is first briefly lowered, and then raised for a longer time. **(B)** The top panel shows the hysteresis in the system: the state shift towards the dominance of the green species occurs faster after the first perturbation than the shift back to the initial stable state after the second perturbation. Introducing commensurate memory (middle and bottom panels) delays the first state shift, thus increasing resistance, and hastens the second state shift, thus mitigating the hysteresis effect and increasing long-term resilience. **(C)** Rapidly alternating opposite

perturbations are applied to the blue species growth rate with a regular frequency. **(D)** Without memory (top), the hysteresis effect leads to a permanent shift towards the green-dominated alternative stable state after a few oscillations. Adding commensurate memory mitigates the hysteresis, thus extending the transitory period (middle), which may generate longstanding oscillations in community composition before the community converges to a stable state (bottom).

recover its initial state if the perturbations follow each other too rapidly. In such circumstances, memoryless communities reach a new stable state faster than the communities with memory, as the latter resist the perturbations for a longer time because of the reduced hysteresis (Fig 3D). This may lead to community dynamics being trapped in long-lasting transient oscillations.

Finally, we analyse the role of stochastic perturbations, which form an essential component of variation in real systems. Under stochastic perturbation (Fig 4A), ecological memory dampens the fluctuations and can significantly delay the shift towards an alternative stable state (Fig 4B). This demonstrates in a more realistic perturbation setting how memory promotes community resistance to perturbation. Our results thus show that memory can enable long-term species coexistence under stochastic or alternating pulse perturbations.

Memory can have unexpected effects on community dynamics when its strength is tuned to bring the system in the vicinity of the tristable region, where the outcome of the dynamics is highly sensitive to initial conditions. Under such conditions, minute changes in memory can push the system over a tipping point towards another attractor, radically changing the outcome (Fig 4C). This illustrates that, beyond introducing inertia into the dynamics and damping perturbations, memory can have non-trivial effects on the system's stability landscape, which we investigate in the next section.

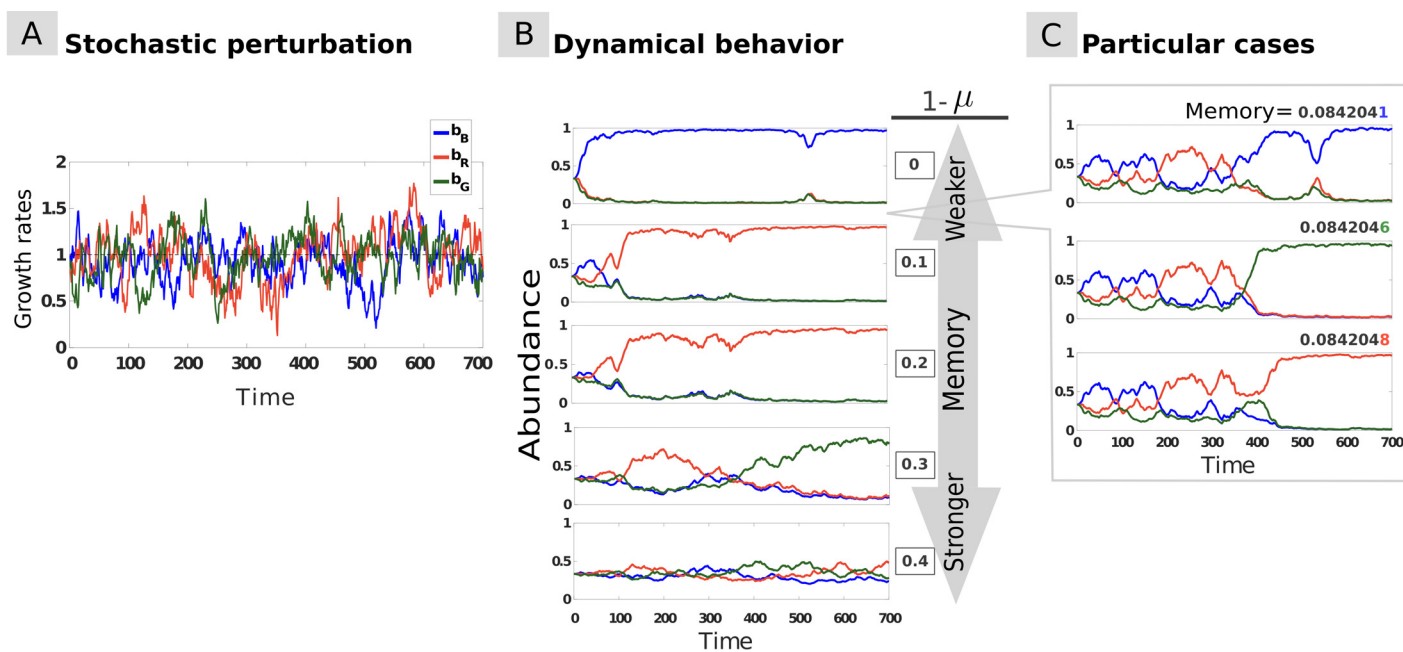

**Fig 4. Stochastic perturbations with commensurate memory effects.** **(A)** Species growth rates $b_i$ vary stochastically through time according to an Ornstein-Uhlenbeck process (see Table in S1 Table). **(B)** Dynamical behavior of the system in response to the stochastic perturbation for equal initial species abundances and varying memory level: in addition to slowing down community dynamics, increasing memory limits the overall variation in species abundances, thus enhancing the overall resistance of the system and promoting species coexistence. **(C)** For some memory strengths, the final state of the system can be sensitive to slight variations in memory, with drastic consequences on community composition.

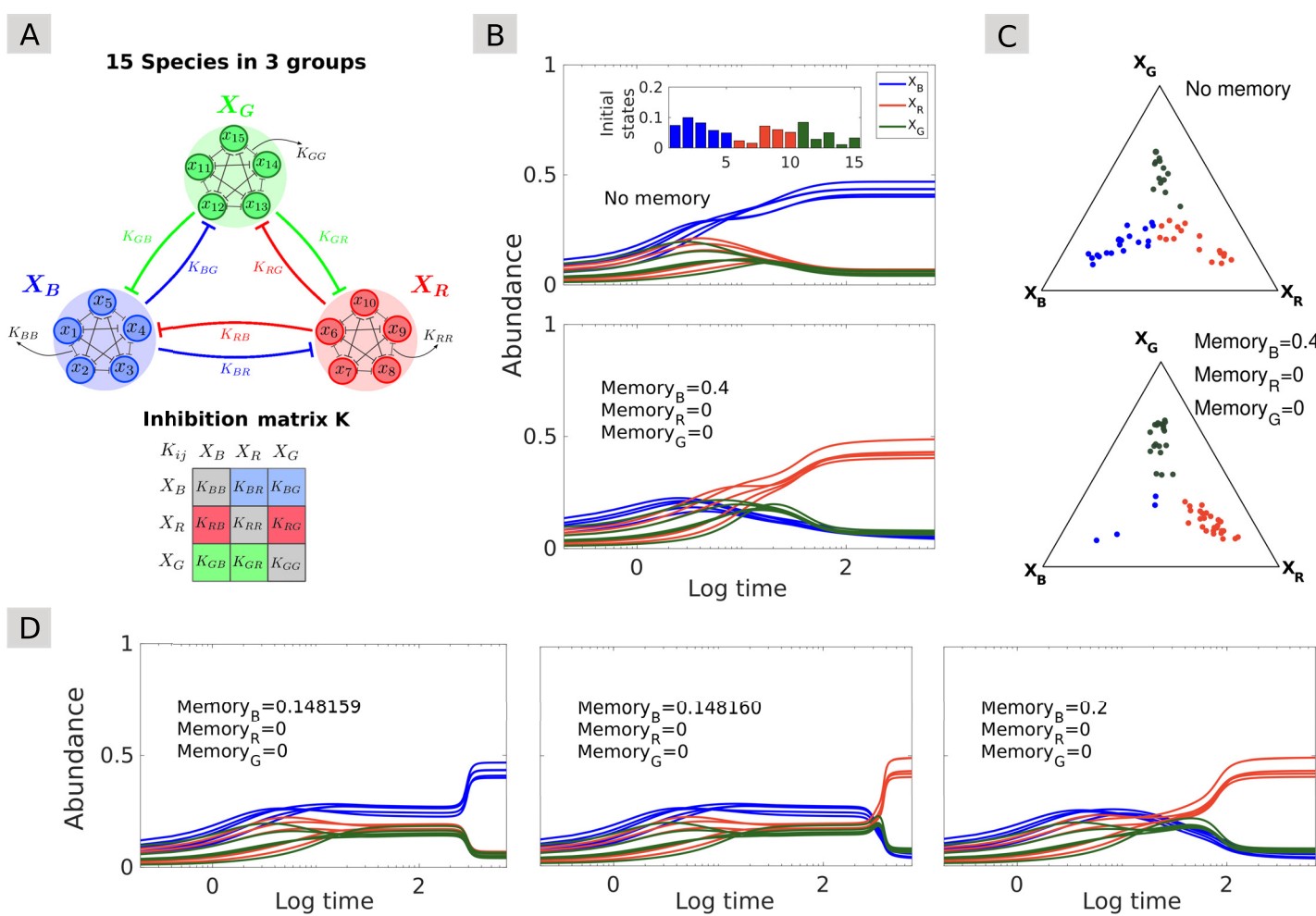

**Fig 5. Impact of incommensurate memory on the community stability landscape: Regime shifts without perturbation. (A)** A 15-species version of Gonze's mutual interaction model (see Gonze et al. [25]). The 15 species form three groups, blue, red, and green, and between-group species interactions are stronger than within-group interactions. The resulting system exhibits three stable states, each dominated by a different group. **(B)** Starting from random initial conditions, the blue group of species dominates the community in the stable state when no memory is present (top). Starting from the same initial conditions, imposing memory on the blue species leads to a temporary rise in abundance, but ultimately another (red) group of species dominates instead (bottom). **(C)** Ternary plots represent the stable state distributions of 50 simulations with random initial conditions and noise in model parameters. Each dot shows, for one simulation at convergence time, the identity of the dominant group (color) and the average relative abundances of the three groups (position in the triangle; see S1 Appendix for details). In the memoryless system (top), the three groups roughly have the same chance of dominating the stable state, whereas imposing memory effects on the blue set of species (bottom) favors stable states where those species are not dominant. **(D)** Setting incommensurate memory in the blue species around the threshold separating two stable states (here, 0.14816) leads to an abrupt regime shift after a long period of subtle, gradual inclines, without changing any model parameters or adding noise.

## Impact on stability landscape

Let us now consider a tristable model equivalent to the one used so far, where the 3-species community is replaced for more generality by a 15-species community structured into three groups through its interaction matrix. Each of these groups represents a set of weakly competing species—*e.g.*, because of cross-feeding interactions that mitigate competition, whereas species belonging to different groups compete more strongly with each other (Fig 5A). We show that adding memory in such a system can change the outcome of the dynamics even in the absence of perturbation. In particular, increasing the strength of (incommensurate) memory in the group that is dominant in the stable state of the memoryless system can lead to its

exclusion from the new stable state (Fig 5B and 5C). This happens because memory shifts the boundary between stable states in the space of initial conditions.

Adding memory in a given species may lead to either a reduction or an increase in its abundance depending on the conditions. Whereas Fig 5B and 5C and S2 Fig illustrate the exclusion of a group of species with higher memory from the stable state in the absence of perturbation, memory may conversely increase the persistence or abundance of a species, as illustrated in S3 (B) Fig in the presence of perturbation. In fact, in the presence of perturbation, tuning memory in a given species may lead to the dominance of any of the species depending on the perturbation and initial conditions. This result holds both in the case of pulse (S3(A) Fig) and stochastic (S3(B) Fig) perturbation.

When setting the memory strength close to the threshold value between two alternative stable states for a given initial condition, we observe long transient dynamics where the community may remain stuck in an unstable state for an extended period of time. After a long period of subtle, gradual changes, the community eventually converges to its stable state in an abrupt regime shift that is not triggered by any perturbation or changes in the model parameters (Fig 5D).

Remarkably, memory can also induce similar long transient dynamics when the system is outside the region of the model's parameter space exhibiting multistability. S4 Fig illustrates how, depending on initial conditions, incommensurate memory may induce long transient states in a community that would, in the absence of memory, rapidly converge to a single stable state irrespective of initial conditions. These transient states are characterized by the dominance of a species or group of species that is not dominant in the stable state. A bifurcation diagram shows that the region of the model's parameter space that exhibits long transient dynamics is next to the multistability region (Fig 6). Memory therefore reveals the "imprint" of alternative stable states that exist in adjacent regions of the parameter space.

## Empirically parameterized model

Our approach to modeling memory effects is general and could be applied to any differential equation system. To study whether the observed memory effects also arise in a more realistic setting, we applied our approach to a gLV system parameterized by Venturelli *et al.* [39] with experimental data from synthetic human gut microbial communities. For demonstration purposes, we restricted ourselves to communities of just two species, which we formed by virtually combining the following bacterial species: *Bacteroides uniformis* (BU), *Bacteroides thetaiotaomicron* (BT), *Clostridium hiranonis* (CH), and *Eubacterium rectale* (ER). We analyzed three different combinations of two species: (i) combining BU and BT, we obtained a *bistable* community converging to distinct stable states depending on initial conditions, similarly to the communities investigated so far (Fig 7A and 7B); (ii) combining CH and ER, we obtained a monostable community exhibiting *stable coexistence*, where neither of the species makes up more than 95% of the total abundance in the stable state (S5(C) Fig); and (iii) combining BT and CH, we obtained a monostable community exhibiting *single species dominance* in the stable state (S5(D) Fig). We compared the results obtained by introducing memory in these empirically parameterized communities with those obtained in a two-species version of the multistable (here bistable) model studied so far, hereafter referred to as Gonze model (S5(A) and S5(B) Fig). We systematically tested a wide range of memory strength values to assess the robustness of our results.

We measured the resistance and resilience of the two bistable community types (i.e., the BU-BT community and the two-species Gonze model) to a pulse perturbation. We measured resistance as the strongest perturbation the community can withstand before shifting to an

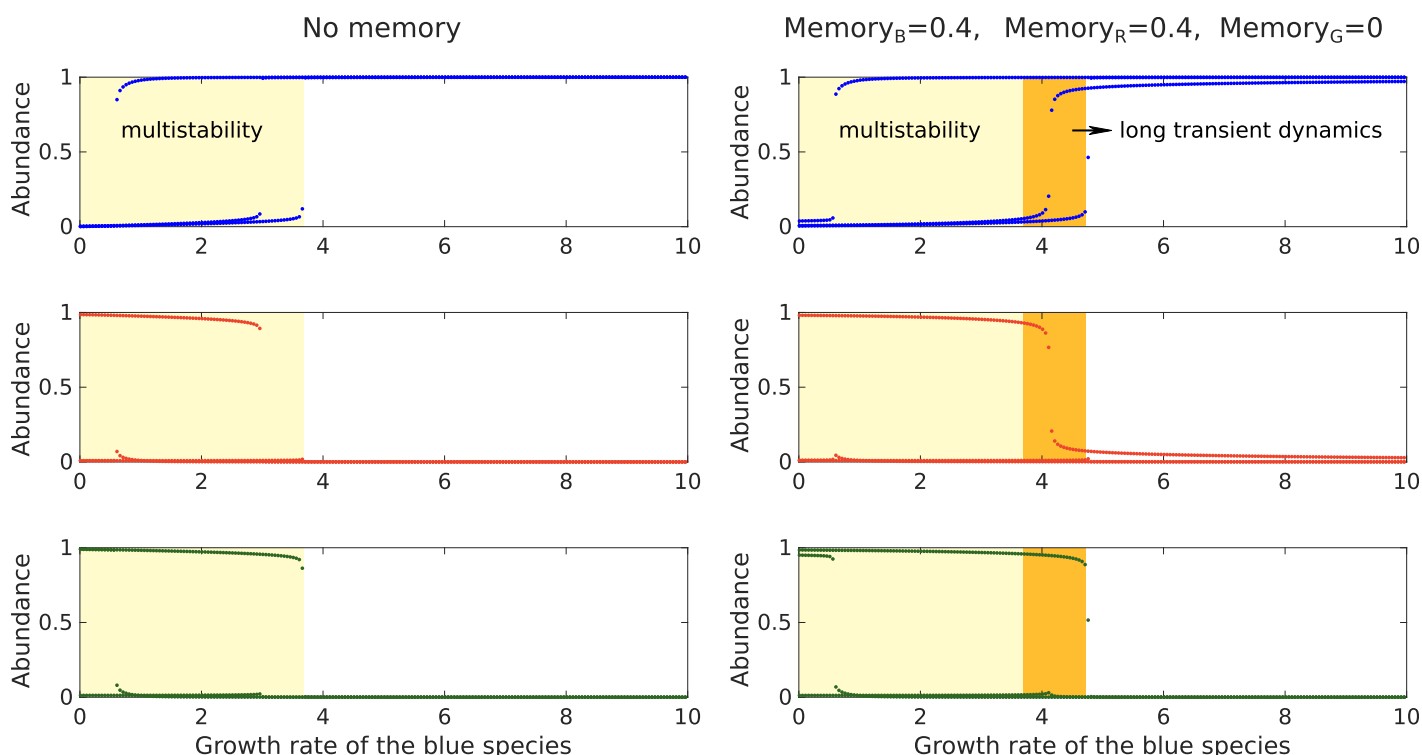

**Fig 6. Memory induces long transient dynamics in a region of the parameter space adjacent to the multistable region.** Bifurcation diagrams for the 3-species Gonze model showing the relative abundances at time 1,000 of the blue, red, and green species as functions of the blue species' growth rate, for three different initial conditions (leading to three distinct curves per plot), and in the absence (left column) or presence (right column) of memory. The light and dark yellow regions exhibit several alternative states for the same parameter values. However, it can be shown analytically that the system only admits a single stable state in the dark yellow region, whereas it admits multiple stable states in the light yellow region. Therefore, alternative states at time 1,000 in the dark yellow region correspond to instances of long transient dynamics, where the system remains stuck near ghost attractor states. See S4 Fig for illustrations of the dynamics in the dark yellow region.

alternative stable state (see Resistance and resilience metrics). In agreement with our previous results, we find that memory consistently increases resistance over the tested range of memory strengths (S6 and S7 Figs). We measured resilience as the recovery time to the stable state after perturbation. In agreement with our previous results, we find that memory hastens the recovery over short time scales (Fig 7E and S8(A) Fig) but slows it down over longer time scales (Fig 7F and S8(B) Fig).

We also measured the convergence time to the stable state in the absence of perturbation under varying memory strength in all community types, that is, in the three empirically parameterized communities as well as in the two-species Gonze model. In general, introducing memory lengthens the convergence time to the stable state, illustrating the inertia induced by memory (S9–S12 Figs), except when memory leads to a change in stable state (in the two bistable communities), in which case introducing memory may influence the convergence time either way (S9 and S10 Figs).

## Discussion

Our understanding of community dynamics heavily relies on mathematical modeling. Dynamical community modeling is a particularly active research area in microbial ecology, where recent studies have proposed numerous mechanistic models of microbial community dynamics exploring the role of interactions, stochasticity, and external factors [2, 25, 48–50].

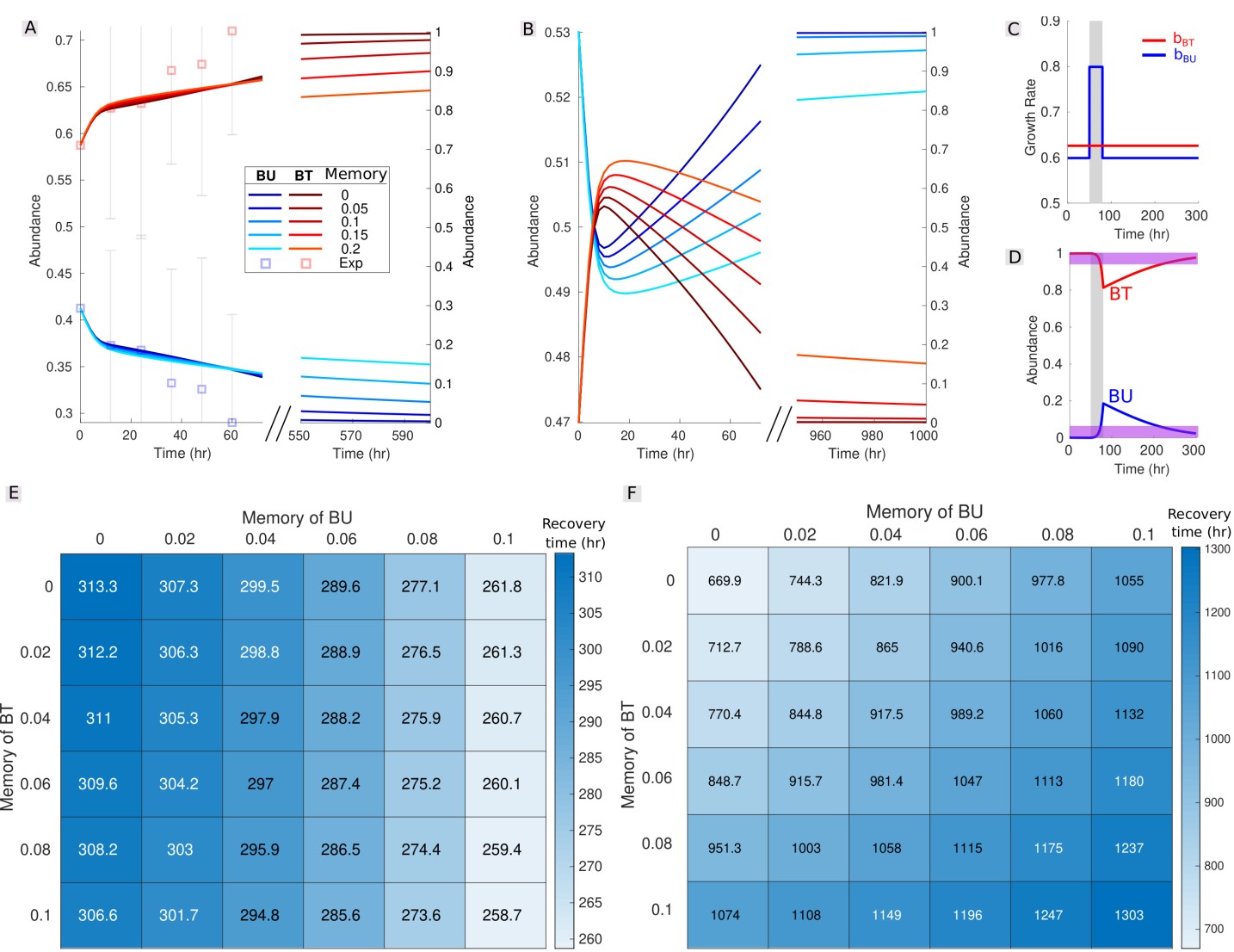

**Fig 7. Impact of memory on the dynamics of an empirically parameterized bistable community. (A-B)** Dynamics of a bistable community composed of *Bacteroides uniformis* (BU) and *Bacteroides thetaiotaomicron* (BT), as simulated from a gLV model fitted to a set of empirical time series by Venturelli *et al.* [39] in which we have introduced memory effects (Eq (2) in the Methods). (A) Dynamics for the same initial conditions as one of the empirical time series used to fit the model. The original empirical data are indicated by squares and (truncated) error bars centered on the squares, representing the mean and standard deviation across biological replicates, respectively. (B) Dynamics for initial conditions leading to the alternative stable state dominated by BU. (A) and (B) illustrate that increasing memory lengthens the convergence time to the stable state, although it may hasten it in initial stages. **(C-D)** For each matrix entry in (E-F), a pulse perturbation is applied to the growth rate of BU (C), which temporarily displaces the community away from its original stable state dominated by BT (D). Convergence is achieved once species abundances reach an arbitrarily set convergence interval (in purple). **(E-F)** Recovery time to stable state after perturbation as a function of memory strength in BU and BT (the upper-left cell corresponds to the memoryless case). Both color and matrix entries indicate the recovery time after a pulse perturbation starting from the BT-dominated stable state, as illustrated in (C-D). **(E)** Recovery times are measured using a loose convergence interval (Bray-Curtis dissimilarity of 0.02 or lower between the community's current state and its initial stable state), thus capturing the early stages of the recovery. Recovery time decreases with increasing memory, i.e., memory effects increase resilience. **(F)** Recovery times are now measured using a much tighter convergence interval (Bray-Curtis dissimilarity of $7e-4$ or lower), thus capturing the later stages of the recovery. Recovery time now decreases with increasing memory, i.e., memory effects decrease resilience. Hence, as for the two-species Gonze model in S8 Fig, the effect of memory on resilience depends on the time scale considered: memory hastens the recovery at first (E) but slows it down further in time (F).

These studies have, however, largely neglected the role of ecological memory despite its potentially remarkable impact on community variation. We have shown here how ecological memory can be incorporated into models of microbial community dynamics and used this modeling tool to demonstrate the role of memory as a key determinant of community

 

dynamics. This has allowed us to expand our understanding of the impact of memory on community response to perturbation, the emergence of alternative community states, long transient dynamics and delayed regime shifts.

Ecological memory is a systemic property that can arise through various mechanisms. First, memory-like delay effects may arise through intracellular mechanisms, such as cell lag phases or inertia in transcriptional regulation, which may be effectively memoryless. In such cases, the dampening effect on the dynamics may be simply modeled by introducing a lag in community dynamics rather than long-term memory effects. In contrast, long-term memory may arise if communities can alter their environment and thus modify environmental parameters in ways that reflect past events, or if organisms exhibit context-specific growth rates that reflect past adaptations [51, 52]. It may also emerge at the population or community level through phenotypic heterogeneity [23, 24], which can be favored by dormancy, spatial structure, or adaptive bet-hedging mechanisms [15–18].

Ecological communities are constantly subject to perturbations arising from external factors, and our analysis focuses in particular on the combined effect of perturbations and memory on community dynamics. Environmental fluctuations through time have a fundamental influence on communities: they may promote species coexistence, increase community diversity [53, 54], contribute to the properties of stable states [43], and in some cases trigger abrupt regime shifts [55]. Our analysis of memory in perturbed communities is particularly relevant to recent studies analysing the response of experimental microbial communities to antibiotic pulse perturbation [1] or the impact of periodic perturbations on the evolution of antimicrobial resistance [40].

The emergence of alternative community states has been recently debated in the microbiome research literature [36, 56]. Gonze et al. [25] demonstrated how pulse perturbations can bring a tristable system to a boundary of the tristability region, which then triggers a transition to an alternative stable state. We have shown how introducing memory into this model can exert additional influence on the resulting dynamics and alter the community's stability landscape. We then assessed the generality and robustness of some of our results by reproducing them in empirically parameterized models. Although we have focused on investigating the effect of memory in deterministic models of biotic interactions, these models could easily be extended to take into account selection by the environment and stochastic drift in species abundances.

We based our modeling of memory on fractional calculus [37], an extensively studied mathematical framework that benefits from well-established mathematical properties (e.g., regarding the existence and uniqueness of solutions). It has a broad range of applications and has already been used to model memory in other fields [57, 58]. In this framework, memory is represented by fractional derivatives and their associated kernel, which determines how quickly the influence of past states fades out (see Methods, Fig 8). Fractional calculus allows introducing memory characterized by a power-law kernel, that is, a power-law decay of the influence of past states on the present state. It is a phenomenological modeling tool that does not assume a specific ecological mechanism for the emergence of memory. It can be regarded as a general approach to the modeling of gradually declining long-term memory, such as the one emerging from phenotypically heterogeneous bacterial populations [15, 16, 23], for instance. Hence, our qualitative results on the influence of long-term memory on community dynamics are likely to hold more generally. One major advantage of fractional calculus is that it can be readily used to introduce memory in any existing dynamical model based on ordinary or partial differential equations, and so can be combined with many types of ecological models. It also allows for fast numerical simulations (see S2 Appendix for details). This makes the

resulting models potentially suitable for simulation-based inference from data, which represents an interesting avenue for future research.

In general, memory adds a certain inertia in community dynamics that tends to damp down fluctuations and can therefore mitigate or prevent more extreme and sudden changes in the system. This may favor species coexistence in the presence of perturbation, and lead to qualitative changes in the dynamics and in community composition under certain conditions. We have clarified in particular how memory influences resistance and resilience to perturbations in communities. An interesting line of research for future work would be to further quantify the influence of memory on the response to perturbation using recently proposed general measures of resilience, such as exit time [59]. Our findings are in agreement with previous studies showing that commensurate fractional derivatives cause intrinsic damping in a system [60, 61], which may delay transitions or shift critical thresholds [38]. Models with incommensurate memory, i.e., with different memory strengths in different species, yield differential equation systems that are mathematically more challenging to analyse, and therefore remain less well understood. Our analyses with incommensurate memory show that memory in a less abundant species tends to have a stronger influence on the overall dynamics than in a dominant one (true in most tested configurations), and that memory in a given species may or may not favor it depending on the context, such as the presence of perturbations.

A result of particular interest is that memory can induce prolonged periods of instability [42], also called long transient dynamics [44]. Specifically, memory appears to favor long "saddle point crawl-by" in regions of the parameter space that exhibit multistability, and to reinforce "ghost attractor states" in neighboring regions of the parameter space [44]. The long transient states we observed could easily be mistaken for genuine stable states over insufficiently long observation times. Long transient dynamics have previously been reported in ecological systems [62] and chemostat experiments [63], and have been linked to stochasticity, multiple time scales, and high dimensionality [44]. Ecological memory provides an alternative, and largely overlooked, mechanism for their emergence. It has been argued that regime shifts may abruptly occur without parameter changes during such long transient dynamics [44], and our results support this view since we have shown that the presence of memory can lead to abrupt regime shifts even in the absence of perturbations.

Several extensions of our model could be considered in future studies to enhance its flexibility and to model memory more generally, such as switching memory on and off along time [38] or applying fractional differential equations with time-varying derivative orders [64]. Alternative approaches have been considered to model ecological memory: a simple approach is to incorporate autocorrelation into the model structure [20], and one could also model memory using distributed delay differential equations (DDE) [65], fractional delay differential equations [66], or memory-dependent integer derivatives [67], which allow for greater flexibility in the shape of kernel functions. However, constructing fractional derivatives analogs of standard models by using kernels other than power-law is mathematically challenging [68], and may fail to meaningfully describe long-term memory effects [69].

The modeling of real systems using models that incorporate memory would benefit from the ability to gather empirical evidence for the presence, strength, and type of memory in the system. Recent literature suggests that it might be possible to empirically detect the presence of memory based on the broad properties of a time series: it has been shown that longitudinal time series of microbial communities may carry detectable signatures of underlying ecological processes [7, 70], and recently Bayesian hierarchical models [11, 19], random forests [12], neural networks [71], and unsupervised Hebbian learning [24] have been proposed to detect signatures of memory in other contexts. Furthermore, specifically designed longitudinal

experiments could be used to characterize memory in real communities. Although direct experimental manipulation of memory in a microbial system is challenging, the manipulation of lag times in E. coli's diauxic shift provides a recent example [72]. We have here incorporated memory and evaluated its impact in a two-species system with experimentally obtained parameters [39], and this approach could be used to provide experimentally testable predictions on community dynamics.

Improving our understanding of the mechanisms underlying community dynamics is a necessity to generate more accurate predictions, and ultimately to develop new techniques for the manipulation of communities. We have combined here theoretical analysis and simulations to explore the various facets of ecological memory and highlight its often overlooked role as a key determinant of community dynamics.

## Methods

In the following, we detail the mathematical aspects of incorporating ecological memory into two non-linear models belonging to the gLV family.

### Model 1: Gonze model

We used, as a starting point, the following memoryless model introduced by Gonze et al. [25] and referred to in this paper as "Gonze model":

$$\frac{dX_i}{dt} = X_i(b_i f_i(\{X_k\}) - k_i X_i),$$

$$f_i(\{X_k\}) = \prod_{\substack{k=1 \\ k \neq i}}^{N} \frac{K_{ik}^n}{K_{ik}^n + X_k^n}. \tag{1}$$

This model describes the dynamics of each microbial species abundance $X_i$ according to its growth rate $b_i$, its death rate $k_i$, and an inhibition term $f_i$ defined by the inter-specific interaction constants $K_{ij}$ and their exponent $n$ (known as the Hill coefficient). $K_{ij}$ represents the inhibition of species $i$ by species $j$: the lower it is, the stronger the inhibition. Although $X_i$ denotes absolute abundances, we represent relative abundances in most figures to ease visual comparison (except in Figs 2, 5B and 5D and in S7–S12 Figs).

**Three-group model.**   We define three sets of species indexed by B (blue), R (red), and G (green). Each species $i$ belongs to exactly one of these three groups. We define the growth rate of each group by the growth vector $\mathbf{b} = [b_B, b_R, b_G]$, where $b_B = \{b_i \mid i \in B\}$, $b_R = \{b_i \mid i \in R\}$, and $b_G = \{b_i \mid i \in G\}$. We also define the inter-specific interaction matrix $\mathbf{K} = \{K_{ij} \mid i, j \in B \text{ or } R \text{ or } G\}$ such that $K_{ij}$ only depends on the group memberships of species $i$ and $j$, plus a small noise term (see Fig 5A and Methods). We first considered a community of three species (*i.e.*, only one species per group), and then a community of 15 species forming three groups with strong inter-group inhibition and weak intra-group inhibition. If the inhibition strength is large enough (small $K_{ij}$), this model can have three coexisting stable states (tristability). This tristable community is dominated by one of the three groups depending on initial species abundances, interaction matrix $\mathbf{K}$, and growth vector $\mathbf{b}$.

**Two-species version.**   In the "Empirically parameterized model" section of the Results, we additionally use a two-species version of this model for the sake of comparison with the empirically parameterized two-species gLV models we introduce in that section (see below). This two-species version exhibits similar properties but is bistable instead of tristable, each stable state being dominated by one of the species.

## Model 2: Empirically parameterized gLV model

To study the impact of memory in a more realistic setting, we examined memory effects in the following dynamic species abundance model of the human gut microbiome, parameterized by Venturelli *et al.* [39] using in vitro interaction experiments:

$$\frac{dX_i}{dt} = X_i \left( b_i + \sum_{j=1}^{N} K_{ij}X_j \right),$$ (2)

where $N$, $b_i$, and $K_{ij}$ indicate the number of species, growth rates, intra-specific interaction coefficients ($i = j$), and inter-specific interaction coefficients ($i \neq j$), respectively. We considered four microbial species, based on the values inferred in [39] for their interaction coefficients and growth rates (see Appendix Figure S27 in [39]): *Bacteroides uniformis* (BU), which is negatively associated with immunological dysfunction [73], *Bacteroides thetaiotaomicron* (BT), which is positively associated with Ulcerative Colitis [74], *Clostridium hiranonis* (CH), and *Eubacterium rectale* (ER), which is positively associated with Type II diabetes [75]. We focused on three two-species communities exhibiting different qualitative behaviors: coexistence (CH and ER, with +/- interaction), dominance (BT and CH, with -/- interaction), and bistability (BU and BT, with -/- interaction).

## Incorporating memory using fractional calculus

Fractional order derivatives have been successfully used to account for memory effects in many disciplines [37, 38, 57]. To introduce memory in the initial models defined by Eqs (1) and (2), we replaced the ordinary first-order time derivatives by fractional derivatives $\mathfrak{D}^{\mu_i}$ (more precisely, Caputo fractional derivatives [76]). These modified models can be expressed, using the simplifying notation $F_i := X_i(b_i f_i(\{X_k\}) - k_i X_i)$ or $F_i := X_i(b_i + \sum_{j=1}^{N} K_{ij}X_j)$, as:

$$\mathfrak{D}^{\mu_i}X_i = F_i, \quad 0 < \mu_i \leq 1, \mu_i \in \mathbb{R}.$$ (3)

Fractional derivatives implicitly introduce a time correlation function, or "memory kernel", which imposes a dependency between the current system state and its past trajectory via a convolution integral (Fig 8A). That is to say, Eq (3) can also be expressed using a first-order derivative, as:

$$\frac{dX_i(t)}{dt} = \int_{t_0}^{t} K_{\mu_i}(t - \tau)F_i(\tau)d\tau.$$ (4)

The memory kernel's decay rate depends on $\mu_i$: the lower the value of $\mu_i$, the slower it will decay (Fig 8). Throughout this article, we quantify memory as $1 - \mu_i$. In the memoryless case ($\mu_i = 1$), the kernel becomes a Dirac delta function, $\delta(t - \tau)$, which results in an ordinary integer-order differential equation. For $0 < \mu_i < 1$, the memory thus introduced can be considered to have a power-law decay in time, a temporal scaling behavior that is common in nature [23, 58, 61, 62, 77]. Indeed, it can be shown that there is a parameter $\mu > 0$ such that the limit $\lim_{t \to \infty} t^{-\mu} K_\mu(t - \tau)$ is a finite constant for fixed $\tau$ [78]. Efficient numerical methods exist to solve fractional-order differential equation systems (see S2 Appendix).

## Resistance and resilience metrics

We define here quantitative metrics of resistance and resilience, which we use to rigorously investigate and summarise the influence of memory on the two-species community models analyzed in the "Empirically parameterized model" section of the results.

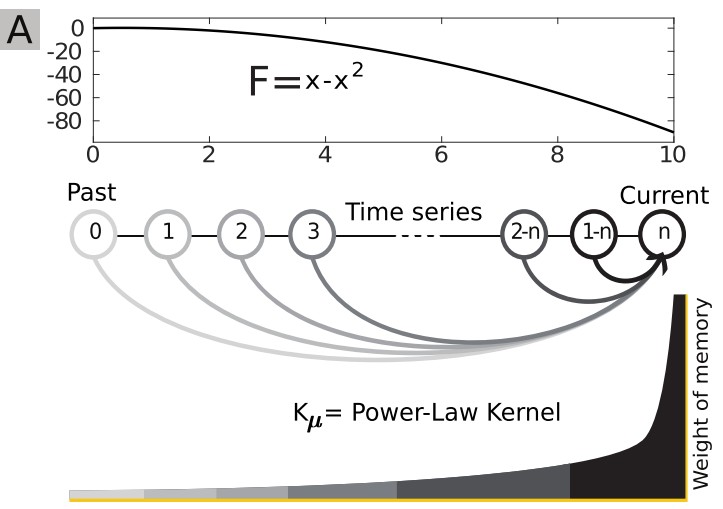
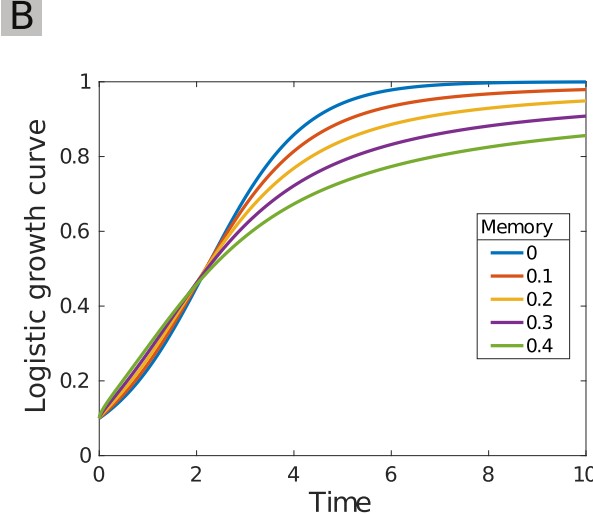

**Fig 8. An intuitive interpretation of the memory introduced by fractional derivatives and of its effect on convergence time for the logistic curve. (A)** The $F$ function in Eq (4) for the standard logistic equation, and a sketch of the memory effects introduced by fractional derivatives on a time series: the weight of past states on the present decreases as a power-law of time. **(B)** Influence of a range of memory strengths on the classic logistic growth curve.

**Convergence time.** To measure the time of convergence to the stable state, we measured through time the Bray-Curtis dissimilarity between the current state of the community, defined by the set of the abundances of all its component species, and its stable state, corresponding to a fixed point of the dynamical system. We measured Bray-Curtis dissimilarity between communities 1 and 2 as $BC = (\sum_{i=1}^{S} |X_{i,1} - X_{i,2}|)/(\sum_{i=1}^{S} (X_{i,1} + X_{i,2}))$, where $X_{i,1}$ and $X_{i,2}$ are the absolute abundances of species $i$ in community 1 and 2 and $S$ is the total number of species. We considered the community to have converged once this dissimilarity was lower than a certain threshold, referred to as the convergence interval. In some cases, we then compared the convergence times obtained with more or less stringent convergence intervals. We used this approach to quantify the convergence time to stability in S8–S12 Figs.

**Resistance.** We measured resistance to perturbation of a multistable community as the strongest perturbation for which the community still recovers to its initial stable state (instead of shifting to an alternative stable state; see Fig 2). We used this approach in S6 and S7 Figs to quantify the resistance of two-species bistable communities to a pulse perturbation in the growth rate of one of the species, starting from the stable state. The strength of the perturbation is defined as the value at which the growth rate is set during the pulse.

**Resilience.** We measured resilience to perturbation as the recovery time to the initial stable state after a perturbation. We used the strongest perturbation for which the community still recovers to its initial stable state. We measured the recovery time as the convergence time to the stable state (see Fig 7C and 7D). We used this approach in Fig 7E and 7F and S8 Fig to quantify the resilience of two-species bistable communities to a pulse perturbation.

## Supporting information

**S1 Appendix. Methodological details for Fig 5C and S1 Fig.**
(PDF)

**S2 Appendix. Numerical simulations.**
(PDF)

**S1 Table. Exact model specifications for the 2, 3, and 15-species Gonze model (Eq (1) in Methods).**
(PDF)

**S2 Table. Exact model specifications for the 2-species model given by Eq (2) in Methods, and for the logistic growth curve in Fig 7.**
(PDF)

**S1 Fig. Memory effects preserve the stable state in randomly structured communities.**
(PDF)

**S2 Fig. Memory in a group of species decreases their relative abundance.**
(PDF)

**S3 Fig. Impact of incommensurate memory in the presence of perturbation.**
(PDF)

**S4 Fig. Memory can induce long transient dynamics even in the absence of multistability.**
(PDF)

**S5 Fig. Memory effects on dynamics for different two-species community types.**
(PDF)

**S6 Fig. Impact of memory on resistance to a pulse perturbation in a two-species community exhibiting bistability between dominance of Bacteroides uniformis (BU) and Bacteroides thetaiotaomicron (BT).**
(PDF)

**S7 Fig. Impact of memory on resistance to a pulse perturbation in the two-species version of Gonze multistable model.**
(PDF)

**S8 Fig. Impact of memory on resilience after a pulse perturbation in the two-species version of Gonze multistable model.**
(PDF)

**S9 Fig. Impact of memory on convergence time in the two-species version of Gonze multistable model.**
(PDF)

**S10 Fig. Impact of memory on convergence time in a two-species community exhibiting bistability between dominance of Bacteroides uniformis (BU) and Bacteroides thetaiotaomicron (BT).**
(PDF)

**S11 Fig. Impact of memory on convergence time in a two-species community exhibiting stable coexistence between Eubacterium rectale (ER) and Clostridium hiranonis (CH).**
(PDF)

**S12 Fig. Impact of memory on convergence time in a two-species community exhibiting stable dominance of Clostridium hiranonis (CH) by Bacteroides thetaiotaomicron (BT).**
(PDF)

## Author Contributions

**Conceptualization:** Moein Khalighi, Guilhem Sommeria-Klein, Didier Gonze, Karoline Faust, Leo Lahti.

**Funding acquisition:** Leo Lahti.

**Investigation:** Moein Khalighi.

**Methodology:** Moein Khalighi.

**Project administration:** Moein Khalighi, Leo Lahti.

**Software:** Moein Khalighi.

**Supervision:** Guilhem Sommeria-Klein, Leo Lahti.

**Visualization:** Moein Khalighi.

**Writing – original draft:** Moein Khalighi.

**Writing – review & editing:** Guilhem Sommeria-Klein, Didier Gonze, Karoline Faust, Leo Lahti.

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
