## [Decision Letter · Decision Letter 0]

24 Nov 2021

Dear MSc. Khalighi,

Thank you very much for submitting your manuscript "Quantifying the impact of ecological memory on the dynamics of interacting communities" for consideration at PLOS Computational Biology.

As with all papers reviewed by the journal, your manuscript was reviewed by members of the editorial board and by several independent reviewers. In light of the reviews (below this email), we would like to invite the resubmission of a significantly-revised version that takes into account the reviewers' comments.

The reviewers are all generally positive about the work, but raise questions that should be addressed before publication. All reviewers discuss concerns related to parameter choice, which therefore merits special attention. We are also very interested in seeing the answer to the questions related to the application of the model to real-world scenarios (another common thread among several of the reviewer comments).

We cannot make any decision about publication until we have seen the revised manuscript and your response to the reviewers' comments. Your revised manuscript is also likely to be sent to reviewers for further evaluation.

Sincerely,

Luis Pedro Coelho

Associate Editor

PLOS Computational Biology

James O'Dwyer

Deputy Editor

PLOS Computational Biology

The reviewers are all generally positive about the work, but raise questions that should be addressed before publication. All reviewers discuss concerns related to parameter choice, which therefore merits special attention. We are also very interested in seeing the answer to the questions related to the application of the model to real-world scenarios (another common thread among several of the reviewer comments).

Reviewer's Responses to Questions

**Comments to the Authors:**

Reviewer #1: The authors study the impact of memory on the dynamics of generalized Lotka-Volterra systems. They show that memory can impact the resilience and resistance of dynamic systems after various perturbations and may impact the resulting stable states.

I think the premise of the manuscript is appealing and it is generally well presented. I agree with the authors that dynamic models can give quantitative insight into the behavior of microbial communities and their results support the general conclusion for the particular system and parametrization that was studied. The manuscript is a bit unclear on what biological system(s) the authors want to simulate which makes it hard to evaluate if the used parametrization has any connection to a real ecosystem or is mostly arbitrary. Even though the manuscript is written excellently some parts are aimed at a very narrow audience and would benefit from some intuitive explanations to make it accessible to readers not familiar with fractional calculus (such as myself).

Suggested major changes

The use of fractional calculus to introduce memory into the generalized Lotka-Volterra model is probably the main contribution in the paper but very little intuition is provide on why that approach is better than alternatives mentioned in lines 21-25. It is also hard to imagine what this particular memory effect actually looks like. So I would have liked to see a figure illustrating the shape of the fractional derivative under various values of μ. For instance, one could have shown the classic logistic growth curve along with varying fractional derivatives or a single impulse curve where one can observe how the impulse propagates in time due to memory. This would make the article accessible to readers without deep knowledge of fractional calculus and give an intuition on what time scales the modeled memory acts.

The authors keep the modeled systems very general which I think is fine. However, at least some part of the manuscript should be focused on a simulation of a system that was parameterized with experimental data to see whether the observed behavior extends to realistic settings. I think the works of Venturelli et. al. could be used here since they estimate gLV parameters in very controlled settings and with extensive data (see https://doi.org/10.15252/msb.20178157 or https://doi.org/10.1038/s41467-021-22938-y, a subset of taxa would be sufficient). In its current form, it is hard to evaluate whether the parameters of the models in Appendix S2 were chosen arbitrarily or selected to produce the particular results observed in the paper. An alternative would be to perform a robustness analysis on those parameters but I would consider this less preferable due to the large numbers of parameters and because behavior arising in only a small part of the parameter space may still be biologically relevant.

Suggested minor changes

It would be great if the authors could expand on which biological mechanism they consider for memory. They mention some examples in lines 209-211. Different memory mechanisms may have vastly different time scales. For instance, acquiring a plasmid may give nearly unlimited memory that exceeds the doubling time but transcriptional regulation is very transient. It is likely that some of those mechanisms are captured better by fractional derivatives than others.

There are some formatting issues in the supplement. The table in Appendix S2 is not numbered and there is some overlapping text on line 492.

I feel like Appendix S1 is pretty essential for what is happening in the manuscript, so I would probably convert it to a methods section in the main text (that may be located at the end of the manuscript).

I don’t have access to a Matlab license so I can not vouch that the provided source code works. I could imagine that providing some more information on how to install dependencies and in which order things should be executed to reproduce the results in the paper may be appreciated by potential readers. The Github repository linked to the Zenodo submission should be mentioned in the paper as well.

Reviewer #2: This is an interesting and well written paper about under-explored topic about the role of “memory” in the community dynamics. I believe this is an important study. Unfortunately, I cannot comment on mathematical implementation of the models because I am not an expert in this field. Several general comments from more biological prospective regarding the manuscript are below.

Major comments.

1) The figure 5C provides a nice example of possible outcomes of community under memoryB=0.4. I believe for the broader audience it would be interesting to see several examples of the “space” of outcomes on a similar figure when all three species have incommensurate memory which is > 0 . (Unless this is computationally too difficult.)

2) How community assembly forces such as the ecological drift and selection relate to memory? Is it possible to speculate to what extent memory facilitates either drift or selection or both?

3) It probably goes beyond the scope of the paper but it would be very interesting to have at least one example of empirical estimates of the memory for a publicly available experimental/observational real or mock community timeseries.

4) Would the interpretations of outcomes of the model be applicable for absolute abundances?

5) It would be useful to have a table or graphical summary in what instances what combinations of values of memory will lead to resistance and when to resilience and recovery of pre-disturbance state.

Minor comments.

1) Statement in the abstract that memory “..thus reducing the system's resilience” seems too strong since later figure 2 shows actually that memory>0 appears to promote resilience.

2) Page 25. Line 510. Fig.3?. Figure number missing

3) Page 24. After line 505. Table number is missing, I assume it is referred to as Appendix S2

4) Double-check if used shades of the green/red colors are color-blind friendly.

Reviewer #3: In the manuscript of “Quantifying the impact of ecological memory on the dynamics of interacting communities”, the authors use the framework of fractional calculus to study how the outcomes of a well-characterized interaction model are affected by gradual increases in ecological memory under varying initial conditions, perturbations, and stochasticity. Results highlight the implications of memory on several key aspects of community dynamics.

The research background is of great interest. The manuscript is well written. However, I have several questions especially on techniques, for example, the quantity of stability, and so on.

Major questions:

1) As shown in SM, the authors started from the memoryless model and then incorporated memory. Could the authors explain the generality of the integral function, Eq. (s) in SM, denoting the effects of memory. Looks like that this definition is crucial. Without different definitions, results could be totally different.

2) Based on the constructed model, there are also different ways of perturbations, as mentioned in Fig. 1, including Pulse, periodic, stochastic, and so on. Could the authors quantify the effects of parameter values of these different types of perturbations on the stability, recovery time, and resilience.

3) Could the authors provide the definition of stability, recovery time, resilience?

So, in general, the research question is very interesting. But it would be nice to provide the definition of metrics clearly and investigate the effects of memory rigorously.

**Have the authors made all data and (if applicable) computational code underlying the findings in their manuscript fully available?**

Reviewer #1: Yes

Reviewer #2: Yes

Reviewer #3: Yes

PLOS authors have the option to publish the peer review history of their article (what does this mean?). If published, this will include your full peer review and any attached files.

Reviewer #1: **Yes: **Christian Diener

Reviewer #2: No

Reviewer #3: No
---

## [Decision Letter · Decision Letter 1]

15 Apr 2022

Dear MSc. Khalighi,

Thank you very much for submitting your manuscript "Quantifying the impact of ecological memory on the dynamics of interacting communities" for consideration at PLOS Computational Biology. As with all papers reviewed by the journal, your manuscript was reviewed by members of the editorial board and by several independent reviewers. The reviewers appreciated the attention to an important topic. Based on the reviews, we are likely to accept this manuscript for publication, providing that you modify the manuscript according to the review recommendations.

While we have not been able to secure the opinion of Reviewer #3, our appreciation is that the authors' response addresses their concerns. Therefore, based on the current opinions, we believe that the major scientific questions have been resolved.

PLOS Computational Biology aims to publish work that addresses biological questions and, thus, we concur with Reviewer #1 that it would enhance the manuscript to present some results from the Section "Empirically parameterized model" as one of the main figures (currently, that section only refers to supplemental figures).

Sincerely,

Luis Pedro Coelho

Associate Editor

PLOS Computational Biology

James O'Dwyer

Deputy Editor

PLOS Computational Biology

[LINK]

While we have not been able to secure the opinion of Reviewer #3, our appreciation is that the authors' response addresses their concerns. Therefore, based on the current opinions, we believe that the major scientific questions have been resolved.

PLOS Computational Biology aims to publish work that addresses biological questions and, thus, we concur with Reviewer #1 that it would enhance the manuscript to present some results from the Section "Empirically parameterized model" as one of the main figures (currently, that section only refers to supplemental figures).

Reviewer's Responses to Questions

**Comments to the Authors:**

Reviewer #1: The authors have expanded their explanation of what phenomena could give rise to the modeled memory effects and now provide a much better illustration and description of how fractional calculus can be used to model the covered memory effects. Additionally, the manuscript now also includes a small study of memory effects within a gLV model derived from bacterial growth curves, thus, providing simulations of memory effects in a somewhat more realistic setting. I think this has already improved the manuscript and I only have some minor suggestions to improve clarity and presentation of the data.

Minor suggestions

The manuscript text and methods are both pretty unclear on how the data from Venturelli et. al. was integrated. Did the authors use the parameters fit in the original manuscript or was the fractional calculus model refit to the time series data? In my opinion this data should have been a figure in the main manuscript as I find simulations in more realistic settings much more interesting than some of the more artificial settings in the other figures. The growth curves shown in Fig. S5 should also be presented alongside with the actual growth curves from the Venturelli manuscript to give an impression how well the simulations match the measured growth curves.

Line 158: I don’t think what was done here constitutes an actual validation since they never compare the simulated memory behavior to measured data. Rather it should state something along the lines of: “To study whether the observed memory effects may arise in a more realistic setting…”, which is what was actually done here.

The heatmaps in figures S5-S8 and others are missing units for the color bars. Is the recovery time expressed in hours, minutes, or seconds?

I think the points brought up by the other reviewer are quite important. If the modeling strategy can not account for selection and drift this should be stated clearly in the manuscript.

Reviewer #2: Authors have addressed all of my questions and suggestions.

In caption to Figure S4, reference to another figure is missing and has two question marks "see Fig. ??"

**Have the authors made all data and (if applicable) computational code underlying the findings in their manuscript fully available?**

Reviewer #1: Yes

Reviewer #2: Yes

PLOS authors have the option to publish the peer review history of their article (what does this mean?). If published, this will include your full peer review and any attached files.

Reviewer #1: **Yes: **Christian Diener

Reviewer #2: **Yes: **Oleksandr Maistrenko

Figure Files:

Data Requirements:

Reproducibility:

References:

---

## [Editor Report · Decision Letter 2]

12 May 2022

Dear MSc. Khalighi,

We are pleased to inform you that your manuscript 'Quantifying the impact of ecological memory on the dynamics of interacting communities' has been provisionally accepted for publication in PLOS Computational Biology.

Best regards,

Luis Pedro Coelho

Associate Editor

PLOS Computational Biology

James O'Dwyer

Deputy Editor

PLOS Computational Biology

---

## [Editor Report · Acceptance letter]

30 May 2022

PCOMPBIOL-D-21-01512R2 

Quantifying the impact of ecological memory on the dynamics of interacting communities

Dear Dr Khalighi,

I am pleased to inform you that your manuscript has been formally accepted for publication in PLOS Computational Biology. Your manuscript is now with our production department and you will be notified of the publication date in due course.

With kind regards,

Anita Estes
